# KINDA-45M : A LARGE-SCALE VIDEO DATASET IMPROVING CONSISTENCY BETWEEN FINE-GRAINED CONDITIONS AND VIDEO CONTENT

## ABSTRACT

As visual generation technologies continue to advance, the scale of video datasets has expanded rapidly, and the quality of these datasets is critical to the performance of video generation models. We argue that temporal splitting, detailed captions, and video quality filtering are three key factors that determine dataset quality. However, existing datasets exhibit various limitations in these areas. To address these challenges, we introduce **Kinda-45M** , a large-scale, high-quality video dataset featuring accurate temporal splitting, detailed captions, and superior video quality. The core of our approach lies in improving the consistency between fine-grained conditions and video content. Specifically, we employ a linear classifier on probability distributions to enhance the accuracy of transition detection, ensuring better temporal consistency. We then provide structured captions for the segmented videos, with an average length of 200 words, to improve text-video alignment. Additionally, we develop a *Video Training Suitability Score (VTSS)* that integrates multiple sub-metrics, allowing us to filter high-quality videos from the original corpus. Finally, we incorporate several metrics into the training process of the generation model, further refining the fine-grained conditions. Our experiments demonstrate the effectiveness of our data processing pipeline and the quality of the proposed Kinda-45M dataset.

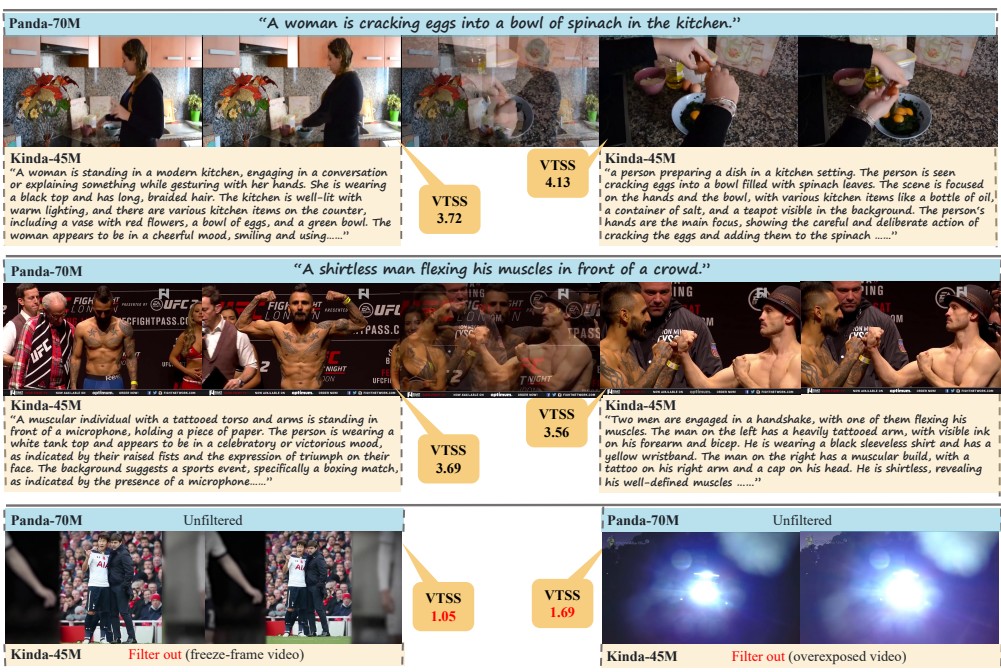

Figure 1: **Comparison between Kinda-45M and Panda-70M.** We propose a large-scale, high-quality dataset that significantly enhances the consistency between multiple conditions and video content. Kinda-45M features more accurate temporal segmentation, more detailed captions, and improved video filtering based on the proposed Video Training Suitability Score (VTSS).

# 1 INTRODUCTION

Generative AI, particularly video generation tasks, has recently garnered significant interest from researchers. These tasks involve generating high-quality videos from textual descriptions or images. A critical factor in the success of these models is the quality of the datasets used for training. Several open-source datasets (*e.g.* Panda-70M(Chen et al., 2024b), MiraData (Ju et al., 2024), OpenVid (Nan et al., 2024), and VidGen (Tan et al., 2024)) have been introduced, each carefully selecting data sources and applying various evaluation metrics for video filtering. Moreover, innovative approaches have been employed in the video captioning process, such as the multi-modal caption model (Chen et al., 2024b) or structured captions (Ju et al., 2024).

Despite the success of the data processing pipelines introduced by previous datasets, we argue that the core challenge lies in establishing accurate and fine-grained conditioning for video data, which is crucial for both reducing the complexity of the training process and improving the quality of the generated outputs. To achieve this, we believe there are three key issues that need to be addressed:

First, the alignment between text and video semantics is essential. Unlike video question answering tasks, where captions are primarily driven by specific question-based details, video generation requires captions that are directly tied to the visual content itself. Due to the infinite granularity of visual signals, this necessitates captions that are rich and detailed. Furthermore, raw video data often contains complex transitions, adding additional challenges in ensuring the accuracy of captions.

Second, the effective evaluation and filtering of low-quality data remains underexplored. Low-quality video data, such as poor visual quality or excessive artificial effects, can impede the training process. However, accurately assessing and filtering such data presents an ongoing challenge. Existing methods typically rely on manually selected quality metrics and heuristic threshold-based filtering, which are often designed for other tasks and may not align with the specific requirements of video generation. As a result, these approaches may not effectively ensure the desired data quality for training.

Third, even with data filtering processes in place, the videos within the dataset still vary in quality, with each video potentially exhibiting different strengths and weaknesses (e.g., one video may have lower clarity but better aesthetic appeal). Training with such heterogeneous data in the same manner may introduce ambiguity for the model, hindering its ability to learn effectively.

To address these issues, we present **Kinda-45M** , a large-scale high-quality video dataset with more accurate video splitting, detailed captions, better data filtering methods and metric conditions. As video content reaches considerable quality, the consistency between fine-grained conditions and video content determines the performance of generation models. We propose a more refined data processing pipeline based on this key insight. Since accurate video splitting leads to better temporal consistency, we first employ a linear classifier on probability distributions to enhance the accuracy of transition detection. Then We generate structured captions for the segmented video clips, with an average length of 200 words, to improve text-video alignment. Sequentially, to prevent the erroneous deletion of high-quality data during filtering, we train a network to predict *Video Training Suitability Score (VTSS)* on human-aligned datasets to model the joint distribution of sub-metrics. This network takes videos and sub-metrics as input, and outputs a single value called *Video Training Suitability Score* as the only metric to filter data. Additionally, we introduce data metrics as extra conditions (*Metric Conditions*) into the generation model during training, helping model distinguish data with different quality and further improving the consistency between fine-grained conditions and video content, which results in better performance and controllability of the generation model.

To further validate Kinda-45M and our data processing pipeline, we train video generation models on different datasets. Both the dataset benchmark and the performance of the video generation model demonstrate the advantage of the Kinda-45M dataset. We perform more ablation studies to demonstrate effectiveness of our data processing pipeline.

Our contributions can be summarized as follows:

- We present a large-scale high-quality dataset called Kinda-45M , with accurate video splitting, detailed captions and higher-quality video content.
- We propose a refined data processing pipeline to further improve the consistency between fine-grained conditions and video content, including transition detection methods, structured caption system, Video Training Suitability Score and metric conditions.

- Comprehensive experiments demonstrate the advantages of Kinda-45M dataset and the effectiveness of our data processing pipeline.

## 2 RELATED WORK

Recent advancements in diffusion models have driven the evolution of image generation models into video generation models. In the field of text-to-video (T2V) generation, significant efforts have been made to develop large-scale T2V models, trained on extensive datasets using traditional U-Net-based diffusion architectures (Zeng et al., 2024; Clark & Jaini, 2024; Ge et al., 2023; Yu et al., 2023; Khachatryan et al., 2023) and Transformer-based (DiT) architectures (Ma et al., 2024; Chen et al., 2023b; Lu et al., 2023; Chen et al., 2024a; Xing et al., 2024). The success of these video generation models heavily depends on the quality of the video-text datasets.

### 2.1 VIDEO DATASETS

While several video datasets (Caba Heilbron et al., 2015; Anne Hendricks et al., 2017; Rohrbach et al., 2015; Zhou et al., 2018; Xu et al., 2016; Wang et al., 2023b; Sanabria et al., 2018; Wang et al., 2023a; Chen et al., 2023a) have been applied to tasks such as action recognition, video understanding, visual question answering (VQA), and video retrieval, there remains an urgent need for a high-quality, open-source dataset specifically tailored for training video generation models, providing rich video-text pairs. Datasets such as YouCook2 (Zhou et al., 2018), VATEX (Wang et al., 2019), and ActivityNet (Caba Heilbron et al., 2015) offer high-quality human caption annotations. Another set of datasets, including Miradata (Ju et al., 2024), VidGen-1M (Tan et al., 2024), and OpenVid-1M (Nan et al., 2024), automatically generate high-quality captions and filter data using manually selected thresholds on multiple dataset metrics.

However, these datasets are insufficient in size to support the training of large models. Datasets, including YT-Temporal-180M (Zellers et al., 2021), HD-VILA-100M (Xue et al., 2022), ACAV (Lee et al., 2021), etc., contain hundreds of millions of video-text pairs, but their captions are automatically generated via speech recognition, leading to subpar quality. Panda70M (Chen et al., 2024b), the largest publicly accessible video-text dataset, has become a popular choice for video generation due to its scale and considerable quality. However, its quality still needs further improvement. Specifically, the captions in Panda-70M often provide simplistic, incomplete descriptions of video content, and the frequent transitions in the training videos can result in semantic inconsistencies, potentially leading to undesired or uncertain transitions in the generated videos.

### 2.2 VIDEO DATA CURATION

As models continue to scale up in size, effective data curation is of paramount importance (Zhou et al., 2023), particularly in the formulation of a well-suited training dataset. This is crucial for enhancing model performance and improving training efficiency during both the pretraining and supervised fine-tuning phases. In the realm of large language models (LLMs), various data curation approaches have been proposed (Xie et al., 2023; Maharana et al., 2023; Tirumala et al., 2023), including optimizations for data quantity, data quality, and domain composition. However, there remains a lack of work exploring data curation strategies in the video domain. Stable Video Diffusion (Blattmann et al., 2023) offers a comprehensive overview of the curation of large-scale video datasets, including techniques such as video clip-

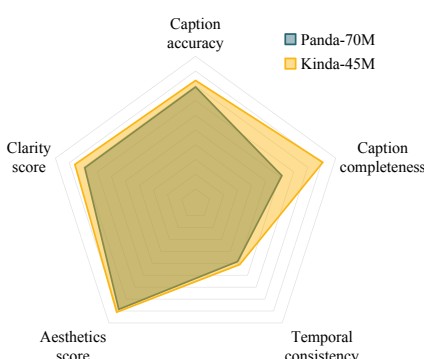

Figure 2: **Quantitative comparison with Panda-70M.** Kinda-45M has a significant improvement in the consistency between fine-grained conditions and video content.

ping, captioning, and filtering. However, the dataset is not open-source. In this study, we propose a novel data processing pipeline for video data and introduce a new video filtering metric. Unlike traditional video quality assessment models (Wu et al., 2023; Zhao et al., 2023; Wu et al., 2022; Sun et al., 2024), which focus primarily on the aesthetic and technical qualities of a video, our approach emphasizes the suitability of videos as training data.

## 3    KINDA-45M DATASET

Kinda-45M is a large-scale high-quality video dataset with accurate video splitting, detailed captions and higher-quality video content. In summary, Kinda-45M contains 45 million video clips with an average duration of 13.75 seconds and a resolution of 720p, each captioned by a text description averaging 202 words in length. We compare Kinda-45M dataset with previous video datasets in Tab. 1. Kinda-45M dataset simultaneously provides a large number of videos (over 10M) and high-quality fine-grained text captions (longer than 200 words), significantly improving the quality of large scale video datasets. Additionally, as shown in Fig. 2, we further compare Kinda-45M with Panda-70M on a series of dataset metrics, such as aesthetic scores and clarity scores, demonstrating a significant improvement in consistency between fine-grained conditions and video content. Since these two datasets come from the same raw datasets, the superiority of Kinda-45M dataset also prove the effectiveness of our data processing pipeline.

Table 1: **Comparison of Kinda-45M and pervious text-video datasets.** Kinda-45M is a video dataset that simultaneously possesses a large number of videos (over 10M) and high-quality fine-grained captions (over 200 words). We propose *structured captions* and *an expert model* (Video Training Suitability Score) for accurate data filtering. "TVL" and "ATL" are abbreviations for "Total Video Length" and "Average Text Length".

| Dataset | #Videos | ATL(words) | TVL(hours) | Text | Filtering | Resolution |
|---|---|---|---|---|---|---|
| LSMDC (Rohrbach et al., 2015) | 118K | 7.0 | 158 | Manual | Sub-metrics | 1080p |
| DiDeMo (Anne Hendricks et al., 2017) | 27K | 8.0 | 87 | Manual | Sub-metrics | - |
| YouCook2 (Zhou et al., 2018) | 14K | 8.8 | 176 | Manual | Sub-metrics | - |
| ActivityNet (Caba Heilbron et al., 2015) | 100K | 13.5 | 849 | Manual | Sub-metrics | - |
| MSR-VTT (Xu et al., 2016) | 10K | 9.3 | 40 | Manual | Sub-metrics | 240p |
| VATEX (Wang et al., 2019) | 41K | 15.2 | ∼115 | Manual | Sub-metrics | - |
| WebVid-10M (Bain et al., 2021) | 10M | 12.0 | 52K | Alt-Text | Sub-metrics | 360p |
| HowTo100M (Miech et al., 2019) | 136M | 4.0 | 135K | ASR | Sub-metrics | 240p |
| HD-VILA-100M (Xue et al., 2022) | 103M | 17.6 | 760.3K | ASR | Sub-metrics | 720p |
| VidGen (Tan et al., 2024) | 1M | 89.3 | - | Generated | Sub-metrics | 720p |
| MiraData (Ju et al., 2024) | 330K | 318.0 | 16K | Generated & Struct | Sub-metrics | 720p |
| Panda-70M (Chen et al., 2024b) | 70M | 13.2 | 167K | Generated | Sub-metrics | 720p |
| **Kinda-45M (Ours)** | 45M | 202.1 | 172K | Generated & Struct | Expert Model | 720p |

## 4    METHOD

As shown in Fig. 3, we propose a refined data processing pipeline for Kinda-45M dataset. Our pipeline aims to further improve the consistency between fine-grained conditions and video content. Our main contributions are shown in the red box of Fig. 3. Specifically, we start from the same raw data with Panda-70M (Chen et al., 2024b) dataset. First, we propose a more accurate and efficient transition detection method for video splitting in section 4.1. Then we caption splitted videos with an average length of 200 words based on our structured caption system in section 4.2. Subsequently, we train a Video Training Suitability Score (VTSS) for data filtering to prevent high-quality data from the erroneous deletion in section 4.3. Finally, we introduce multiple data sub-metrics as *Metric Conditions* into the generation model to enrich the fine-grained conditions in section 4.4.

### 4.1    VIDEO SPLITTING

Splitting videos into temporal segments is crucial for creating video generation datasets. Transition-free video data enable more accurate alignment between text and video, while reducing the difficulty of model training and improving the temporal consistency of generated results. Current video splitting methods (Castellano) typically detect transitions based on changes in image features between consecutive frames, relying on manually adjusted thresholds as criteria, but often overlook temporal information. As a result, these methods struggle to distinguish between gradual transitions and fast-motion scenes, leading to missed detections in the former and incorrect detections in the latter.

To address the above issues, we first propose a Color-Struct SVM (CSS) module that adopting a learning-based approach for more accurate detection of changes between frames compared to threshold-basd method. Then we leverage temporal smoothing and statistical features to differentiate between gradual transitions and fast-motion scenes.

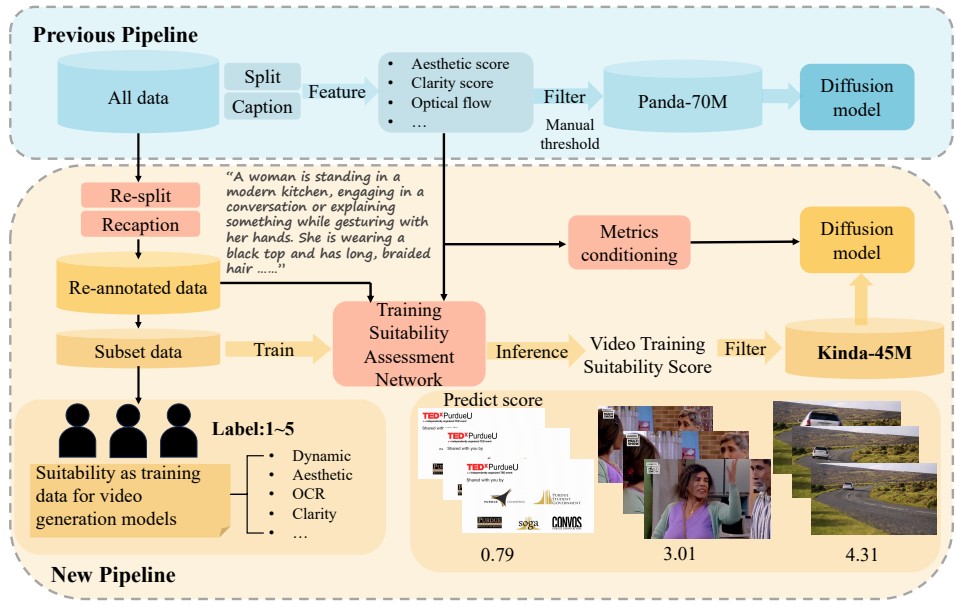

Figure 3: **The proposed data processing pipeline.** Compared with previous pipeline, we propose better splitting methods, structured caption system, training suitability assessment network and fine-grained conditioning in red box, improving the consistency between conditions and video content.

We assume that transitions occur with a low probability at any given moment in the video. We treat image pairs from the same video source as negative examples and pairs from different video sources as positive examples. We select BGR histogram correlation to measure color distance and Canny Luminance SSIM to measure structural distance, which together measure inter-frame changes. For images $I_i$ and $I_j$, the color distance $d_{color}$ and structural distance $d_{struct}$ are defined as follows:

$$H_i = \text{Histogram}(bgr(I_i)) \tag{1}$$

$$d_{color}(H_i, H_j) = \frac{\sum_p (H_i(p) - \bar{H}_i)(H_j(p) - \bar{H}_j)}{\sqrt{\sum_p (H_i(p) - \bar{H}_i)^2 (H_j(p) - \bar{H}_j)^2}} \tag{2}$$

$$E_i = \max(\text{Gray}(I_i), \text{Canny}(\text{Gray}(I_i))) \tag{3}$$

$$d_{struct}(E_i, E_j) = \text{SSIM}(E_i, E_j) \tag{4}$$

Then an SVM classifier is employed, using color distance $d_{color}$ and structural distance $d_{struct}$ as the relevant input features; see Eq. 1, Eq. 2, Eq. 3, Eq. 4 . Regarding temporal information, we hypothesize that video changes are relatively stable over time. By estimating a Gaussian distribution of changes from past frames, if the current frame's change exceeds the $3\sigma$ confidence interval, we consider it a significant transition. This method enhances the differentiation between gradual transitions and fast-motion scenes without increasing computational load. Extensive experiments demonstrate the effectiveness of the transition detection method in A.1.

## 4.2 VIDEO CAPTIONING

Detailed captions usually lead to better text-video consistency, which largely determines the granularity of semantic responses. To obtain more detailed captions, we propose a structured caption system, which consists of: (1) the subject, (2) actions of the subject, (3) the environment in which the subject is located, (4) the visual language including style, composition, lighting, etc. (5) the camera language including camera movement, angles, focal length, shot sizes, etc. (6) world knowledge. We generate these aspects separately, and merge them as the final caption.

Similar to previous works (Chen et al., 2024b; Tan et al., 2024; Ju et al., 2024), we first collect a caption dataset by using GPT-4V (OpenAI, 2023) to generate video captions based on our structured system. We then fine-tune a caption model based on LLaVA (Liu et al., 2023) for the entire dataset. Our experiments during fine-tuning show that training the vision encoder improves the accuracy of the caption. And a high-resolution vision encoder helps the caption model capture video details better. To alleviate the computational burden caused by high-resolution inputs, we perform average pooling with a 2x2 kernel on

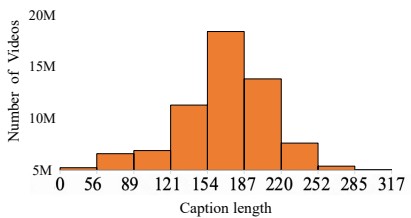

Figure 4: **Distribution of the caption length** (in words) in Kinda-45M dataset.

the spatial dimensions of the tokens, ensuring minimal information loss. Notably, we adopt a mixed training strategy involving both static images and dynamic videos, enabling the model to concurrently learn visual understanding in both static and dynamic scenarios. This also enhances data diversity, alleviating the issue of insufficient training samples when solely relying on video data.

When utilizing the caption model to describe videos, a structured caption system often generates longer captions (over 300 words). Different from MiraData (Ju et al., 2024), we limit the caption length to around 200 words. Because the information entropy of the video is finite, and longer captions may repeat mentioned concepts frequently, making it harder for the generation model to extract key information. Finally, we run our captioner on the whole dataset, and the distribution of caption lengths is shown in Fig. 4. Furthermore, we evaluate the quality of captions with caption accuracy and completeness. As shown in Fig. 2 and Tab. 1, our structured caption system significantly improve the quality of captions, providing better text-video consistency.

### 4.3 DATA FILTERING

In the large-scale raw dataset, the quality of video content vary significantly. When the performance of the generation model is built upon videos with considerable content quality, it is necessary and crucial to filter out low-quality data and remain high-quality data accurately. Traditional methods often use various sub-metrics to evaluate video quality and then manually set thresholds to filter the desired data. Since these sub-metrics are not completely orthogonal with each other, the video quality is actually a joint distribution of all sub-metrics, which means these thresholds should have implicit constraints with each other. However, existing methods neglect the joint distribution of sub-metrics, resulting in inaccurate thresholds. Meanwhile, since multiple thresholds need to be set, the cumulative effect of inaccurate threshold lead to larger deviations during filtering. Therefore, not only low-quality videos are not correctly filtered out as shown in Fig. 1, but also high-quality videos are mistakenly deleted as shown in Fig. 5.

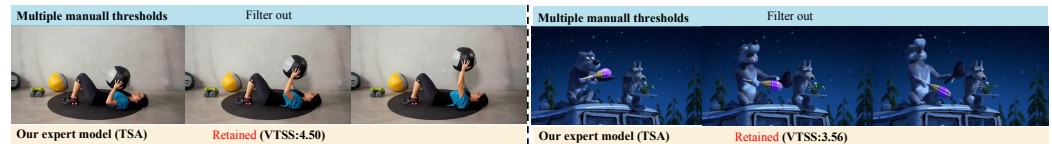

Figure 5: **The deleted high-quality data by inaccurate multiple manual thresholds.**

To address this issue, we propose a *Training Suitability Assessment Network* to model the joint distribution of sub-metrics. This network takes videos and sub-metrics as input, and outputs a single value called *Video Training Suitability Score (VTSS)* as the only metric to filter data. This score reflects whether a video is suitable for training purposes. Specifically, we first collect the training set from human evaluation based on a new criteria. Then we train the *Training Suitability Assessment Network* and employ it to calculate VTSS for all videos. Finally, we set a single threshold for VTSS based on its distribution to filter desired data.

#### 4.3.1 NEW CRITERIA AND HUMAN EVALUATION

We have defined a new annotation criterion that assigns a score reflecting whether a video is suitable as training data for video generation models. This criterion primarily considers the following aspects of video quality: **Dynamic Quality**: A high-quality video should exhibit good dynamics, which are evaluated based on two factors: the extent of subject movement and the temporal stability

of the motion. The motion area in the video should cover more than 30% of the frame; otherwise, the score of the video will be decreased for insufficient dynamics. Temporal stability considers the camera movement; non-professional videographers often produce videos with irregular and significant shaking. We decrease the scores of such videos to distinguish them from professional works. **Static Quality**: Each frame of a high-quality video should have rich subject details, reasonable composition, aesthetic appeal, clear and distinct subjects, and saturated colors. Although this metric may involve some subjectivity, it is crucial for assessing the overall visual quality. **Video Naturalness**: We prefer videos that are natural and unprocessed. Special effects, transitions, subtitles, and logos can introduce biases in the video's original distribution, making it harder for generation models to learn. Additionally, we consider the safety of the video content, rejecting videos with political, terrorist, violent, pornographic, gory, or otherwise disturbing content. In order to reduce the bias between the labeled scores and the true scores, each video is labeled by 8 experts and subjected to a bias elimination process, as described in the App A.2.

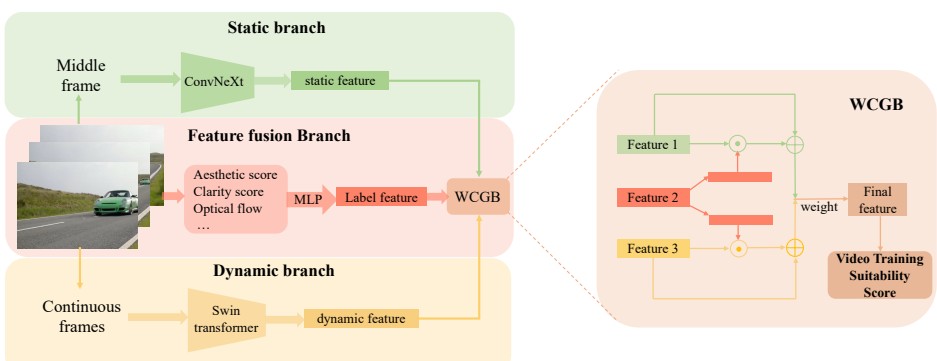

Figure 7: **The pipeline of Training Suitability Assessment Network.**

### 4.3.2 TRAINING SUITABILITY ASSESSMENT NETWORK

As shown in Fig. 7, we propose a *Training Suitability Assessment Network*, which takes videos and sub-metrics as input, and outputs a single value called *Video Training Suitability Score (VTSS)*. Corresponding to the aforementioned annotation criteria, our network is divided into dynamic and static branches. Additionally, we retain various data labels from traditional data filtering strategies and pass this extra information to the network model as a new branch. For the features of different branches, the 3D Swin Transformer is used as the backbone for the dynamic branch, while the ConvNext network serves as the backbone for the static branch. To

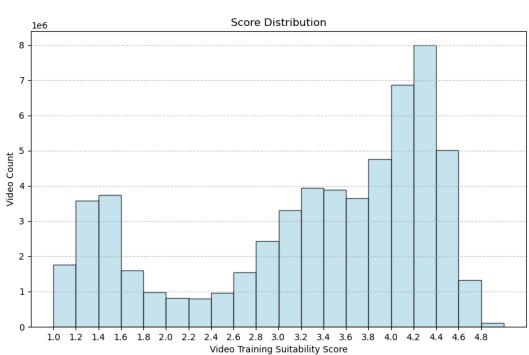

Figure 6: **The distribution of Video Training Suitability Score.**

integrate the features from different branches, we propose a Weight Cross-Gating Block (WCGB) to incorporate the information from the label branch into the other two branches. Since the label branch inherently reflects various characteristics of the video, which are related to both dynamic and static features, we use label features to enhance the dynamic and static features. Given that different video labels focus on dynamic and static aspects to varying degrees, we learn a fusion weight to adjust the proportion of label features integrated with the two types of video features.

After training *Training Suitability Assessment Network* on the human-aligned dataset, we employ it to predict *Video Training Suitability Score (VTSS)* for all videos, and obtain the score distribution as shown in Fig. 6. Since the VTSS distribution can roughly be divided into two Gaussian distributions, we simply chose the decomposition value 2.5 as the VTSS threshold. Based on this threshold, we filtered out a dataset containing a total of 45 million video clips with corresponding captions. And we name the dataset as Kinda-45M , which is the final dataset we present.

### 4.4 METRICS CONDITIONING

In previous pipelines, data metrics are simply used for data filtering. Meanwhile, the quality of the filtered data still varied, making it difficult for the model to distinguish between high-quality and low-quality data. To address this issue, we propose a more fine-grained conditioning method to incorporate quality information of different videos into the generation model during training, leading to better consistency between conditions and video content. During inference, this method also enables fine-grained control over the generated videos.

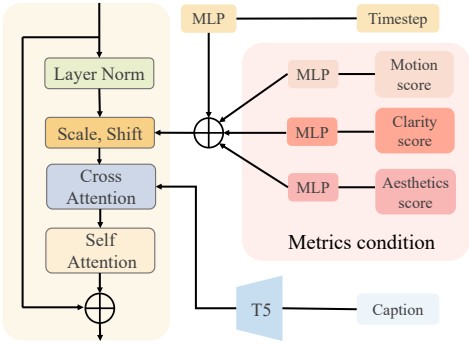

Specifically, during video diffusion training, we first encode data metrics such as motion score, aesthetic score, and clarity score into frequency embeddings. Subsequently, frequency embeddings are passed through an MLP to obtain multiple embeddings, which are then directly added to the timestep embeddings and incorporated into the transformer block using Adaptive Layer Normalization (AdaLN). This method has two main advantages. First, it does not increase the computational load of the diffusion model. Second, compared to adding conditions in captions like Open-sora (Zangwei et al., 2024), it allows for more precise control by being more sensitive to numerical scores, and posses a stronger ability to decouple control over different metrics. During the inference stage, we can set different feature scores, such as setting all scores to the highest value, to generate high-quality videos.

Figure 8: **The pipeline of metrics conditions.**

## 5 EXPERIMENTS

### 5.1 EXPERIMENT SETTING

To validate he superiority of Kinda-45M dataset and the effectiveness of our data processing pipeline, we train the same generation model from scratch on different datasets for comparison. Our text-to-video base model is based on a 3D attention-like Sora structure (Brooks et al., 2024), and the VAE employs a causal convolution-based 3D VAE. Since the training was done from scratch, we set the video duration to 2 seconds and the resolution to 256x256 for faster convergence. All models are trained on their respective datasets passing through 140M data samples in total. To evaluate the performance of generation models, we conduct a comprehensive evaluation on the public benchmark VBench (Huang et al., 2023). Due to the domain gap between the captions provided by VBench and training set, we performed prompt expansion on the captions in VBench.

### 5.2 QUANTITATIVE RESULTS

Table 2: **Quantitative results of text-to-video generation.** We compare the performance of generation models trained on different datasets with VBench. The generation model trained on Kinda-45M surpasses other models on both **quality score** and **semantic score**, with the highest **total score**.

| VBench | Aesthetic Quality | Scene | Subject Consistency | Background Consistency | Temporal Flickering | Motion Smoothness | Dynamic Degree | Imaging Quality | Object Class | Multiple Objects |
|---|---|---|---|---|---|---|---|---|---|---|
| Panda-70M | 0.3988 | 0.1106 | **0.8584** | **0.9435** | 0.9576 | 0.9742 | 0.7722 | 0.4250 | 0.3017 | 0.0223 |
| Kinda-all | 0.4808 | 0.2105 | **0.9335** | **0.9668** | **0.9857** | 0.9855 | 0.4222 | 0.5535 | 0.5453 | 0.1154 |
| Kinda-46M (manual threshold) | 0.4683 | 0.2135 | 0.9388 | 0.9664 | 0.9810 | **0.9870** | 0.4028 | 0.5422 | 0.4858 | 0.1099 |
| Kinda-45M | 0.4832 | 0.1994 | 0.9245 | 0.9613 | 0.9766 | 0.9851 | 0.5750 | **0.5585** | 0.4739 | 0.1145 |
| Kinda-all (condition) | 0.5272 | **0.3211** | 0.9162 | 0.9514 | 0.9210 | 0.9718 | **0.9833** | 0.5316 | **0.7734** | 0.2492 |
| Kinda-45M (condition) | **0.5318** | 0.3163 | 0.9222 | 0.9554 | 0.9246 | 0.9768 | 0.9194 | 0.5344 | 0.7794 | **0.2953** |

| VBench | Human Action | Color | Spatial Relationship | Temporal Style | Appearance Style | Overall Consistency | Quality Score | Semantic Score | Total Score |
|---|---|---|---|---|---|---|---|---|---|
| panda-70M | 0.2400 | 0.5942 | 0.0482 | 0.1281 | 0.2014 | 0.1404 | 0.7343 | 0.3093 | 0.6493 |
| Kinda-all | 0.5180 | 0.8958 | 0.2168 | 0.1630 | 0.1971 | 0.1881 | 0.7758 | 0.4668 | 0.7140 |
| Kinda-46M (manual threshold) | 0.4700 | 0.9128 | 0.1978 | 0.1589 | 0.2003 | 0.1893 | 0.7704 | 0.4548 | 0.7073 |
| Kinda-45M | 0.4880 | **0.9172** | 0.1923 | 0.1571 | 0.1960 | 0.1850 | 0.7819 | 0.4504 | 0.7156 |
| Kinda-all (condition) | **0.8280** | 0.9106 | 0.2434 | 0.2039 | **0.2019** | 0.2277 | 0.7823 | 0.5874 | 0.7433 |
| Kinda-45M (condition) | 0.8080 | 0.8960 | **0.2689** | **0.2045** | 0.2009 | **0.2279** | **0.7846** | **0.5915** | **0.7460** |

As shown in Tab. 2, we comprehensively evaluate models trained on Panda-70M and our dataset at the same step. The generation model trained on Kinda-45M surpasses other models on both **quality score** and **semantic score**, with the highest **total score**. Furthermore, we visualize the VBench metrics comparison in Fig. 9. Kinda-45M significantly improves the generation model's performance on aesthetic quality, object class, multi-objects, human action, and color.

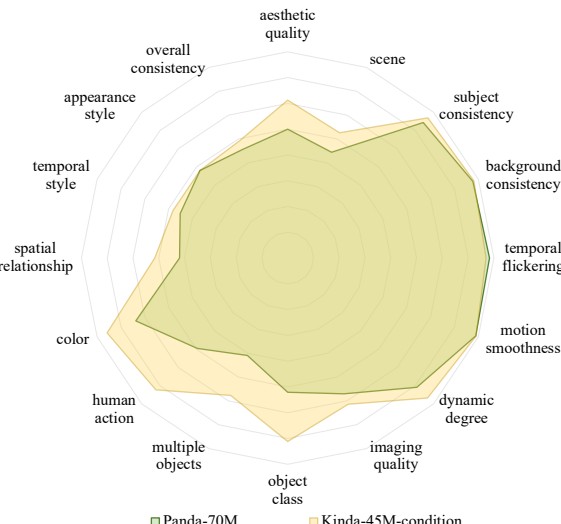

Figure 9: **Visualization of quantitative results of text-to-video generation.** Kinda-45M significantly improves the generation model's performance on aesthetic quality, object class, multi-objects, human action, and color.

## 5.3 QUALITATIVE RESULTS

We visualize the generated videos on VBench's prompts in Fig. 10. The generation model achieve the optimal performance on Kinda-45M , with both the best video quality and text-video consistency. Kinda-45M outperform the larger Panda-70M dataset with only 45M data, indicating that our data quality far exceeds that of Panda-70M. See A.5 for more video generation results.

## 5.4 ABLATION EXPERIMENTS

We conduct extensive ablation experiments to demonstrate the superiority of our dataset and the entire pipeline. Specifically, we performed ablation experiments on different data processing and training strategies, divided into the following groups: (1) **Panda-70M**: baseline. (2) **Kinda-all**: All 58M data after video splitting and captioning. (3) **Kinda-46M**: manually filtered data from Kinda-all using multiple thresholds. (4) **Kinda-45M**: filtered dataset from Kinda-all using VTSS. (5) **Kinda-all-condition**: Kinda-all with metrics conditions. (6) **Kinda-45M-condition**: Kinda-45M with metrics conditions.

**Data Processing.** Comparing the results of training from Panda-70M and Kinda-all in Tab. 2 and Fig. 10, we find that Kinda-all produce better results, especially in temporal quality, such as subject consistency, background consistency and temporal flickering. This indicates that our newly proposed re-splitting algorithm can more accurately segment transitions, reducing semantic inconsistencies between video segments. Additionally, our recaptioning algorithm provided more detailed video descriptions, making it easier for the model to learn the relationship between visual and textual information. To further demonstrate the superiority of our splitting and captioning methods, we conducted extensive comparative experiments, detailed in the App. A.1.

**Data Filtering.** Comparing the results of training from Kinda-all&Kinda-45M and Kinda-all-condition&Kinda-45M-condition, we find that the results from the latter one perform better than that from the former datasets. This indicates that filtering out low-quality data and retaining high-quality data are necessary to prevent the model from learning biased distributions from low-quality data. In addition, comparing the results of training from Kinda-46&Kinda-45M, it can be concluded that our filtering method based on single VTSS results in better filtering performance, when more

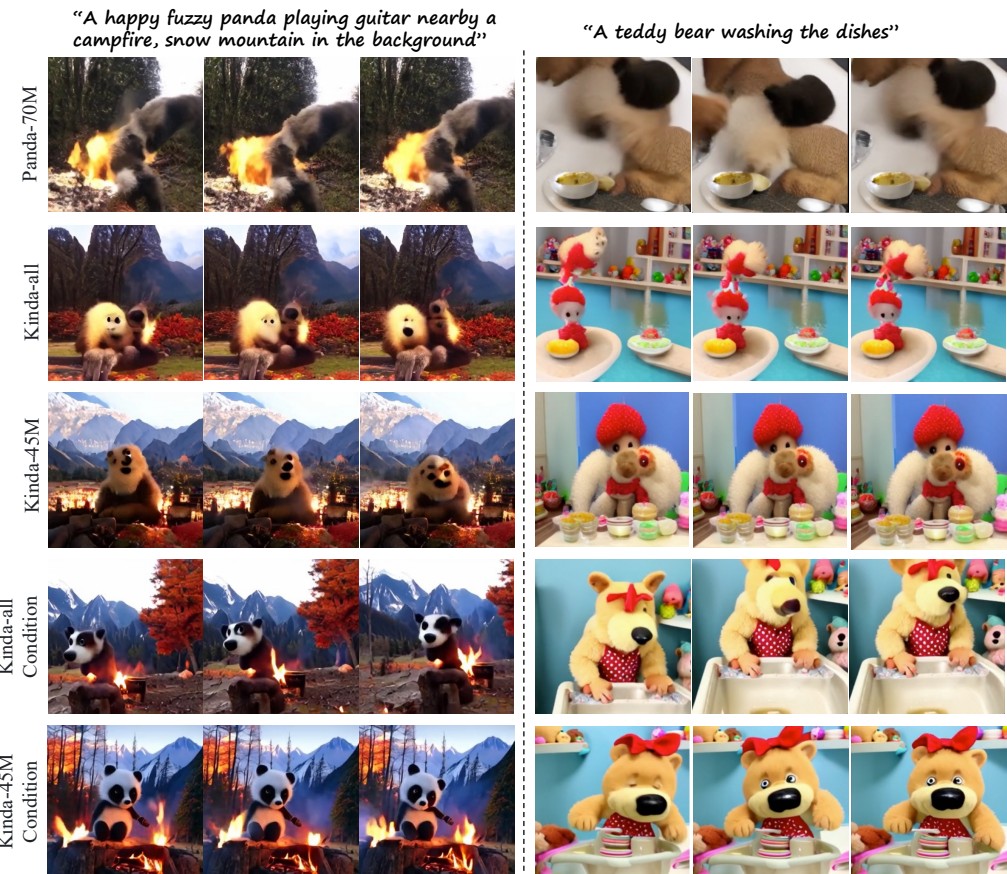

Figure 10: **Qualitative results of text-to-video generation.** We train the same generation model from scratch on different datasets for comparison. The generation model achieve the optimal performance on Kinda-45M , with both the best video quality and text-video consistency.

high-quality data and less low-quality data being retained. Extensive ablation experiments of *Training Suitability Assessment Network* are conducted in the App. A.3.

**Metrics conditions.** Comparing the results of training from Kinda-45M&Kinda-45M-condition, the generation model shows significant improvements in video quality, when metrics conditions are injected into it. This indicates that guiding model training using sub-metrics is necessary, as it helps the model implicitly model the importance of different data. In addition, we compare our AdaLN-based injection method with text-encoder based method (Zangwei et al., 2024) in App. A.4 Fig. 13. It can be discovered that our injection method has more precise control and stronger ability to decouple control over different metrics, when the style of videos transfer with the motion score.

## 6 CONCLUSION

In this paper, we present a large-scale high-quality dataset called Kinda-45M , with accurate video splitting, detailed captions and higher quality video content. Kinda-45M dataset is currently the only video dataset that simultaneously possesses a large number of videos (over 10M) and high-quality fine-grained text captions (longer than 200 words), significantly improving the quality of large scale video datasets. Additionally, we propose a refined data processing pipeline to further improve the consistency between fine-grained conditions and video content, including better transition detection method, structured caption system, and data filtering method and fine-grained conditioning.

**Limitations.** Despite all the strength above, Kinda-45M is still insufficient to support the training of an extremely large video generation model with over 1B parameters. A larger-scale datasets need to be further collected and processed. Meanwhile, the performance, generalization, and scaling laws of generation models on high-quality datasets need further exploration.

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

# A  APPENDIX

## A.1  EFFECTIVENESS OF VIDEO SPLITTING METHODS

You may include other additional sections here. To validate the accuracy and efficiency of our proposed Color-Struct SVM (CSS) for scene transition detection, we conduct the following experiments.

We annotate transitions in 10,000 video clips, creating a test set (approximately half of the videos contain transitions). We then apply our proposed method and an open-source method to detect transitions in the test set, recording the precision and recall of the detections. The open-source method is primarily based on pyscenedetectCastellano, and we test two versions: one that detects transitions based solely on HSL (Hue, Saturation, Lightness) and another that uses both HSL and edge detection. The experimental results are shown in the table below. It can be observed that our transition detection algorithm outperforms the two pyscenedetect-based methods in terms of both precision and recall (Tab. 3). Notably, our algorithm achieves a high recall rate, indicating that it rarely misses transitions in videos.

Table 3: **Transitions Detection Metrics for Different Methods**

| Method | Accuracy | Recall | Precision |
|---|---|---|---|
| Pydetect(hsl) | 0.4421 | 0.3096 | 0.5920 |
| Pydetect(hsl+edge) | 0.4574 | 0.4146 | 0.5854 |
| Ours | **0.7741** | **0.9395** | **0.7547** |

On the other hand, we compare the runtime efficiency of our method with that of the open-source algorithms. We record the CPU runtime of our algorithm and other open-source algorithms at different resolutions, with the experimental results shown in Tab 4. We find that at a resolution of 256x256, our method performs comparably to other methods. However, as the video resolution increases, our method becomes significantly faster than the other methods (Fig 11).

Table 4: **Time Consumption for Different Resolutions and Methods(ms)**

| Resolution | Our Method | Pydetect(hsl) | Pydetect(hsl+edge) |
|---|---|---|---|
| $256^2$ | 1.42 | **0.68** | 2.50 |
| $512^2$ | **2.45** | 2.63 | 8.82 |
| 720p | **6.15** | 10.73 | 30.57 |
| 1080p | **12.26** | 26.16 | 70.11 |
| 4k | **41.98** | 102.55 | 267.18 |

## A.2  ELIMINATION OF DEVIATIONS BETWEEN TRUE SCORES AND LABELED SCORES

After establishing the criteria, we randomly sample a batch of data and have it annotated by trained experts, with each video being scored by eight experts on a scale of 1 to 5. To ensure that the annotations closely reflect the true suitability scores, we need to address two types of errors: **Individual Preference Bias**: As shown in the Fig. 12, we visualize the violin plots of scores given by different experts. The expert on the left tends to give lower scores, while the expert on the right tends to give higher scores. These individual preferences can cause the final scores of some videos to be lower or higher than their actual values. Therefore, we standardize the scores of each expert and then scaled them using the mean and variance of the overall scores to eliminate the bias introduced by different experts. From the figure, it can be seen that the scores processed through our normalization and rescaling methods align more closely with the overall score distribution. **Label Fluctuation Bias**: As shown in the Fig. 12, each video is annotated by eight experts, and different experts may assign different scores due to varying interpretations of the criteria. This leads to label fluctuations. We use the mean score to reduce the error caused by these fluctuations.

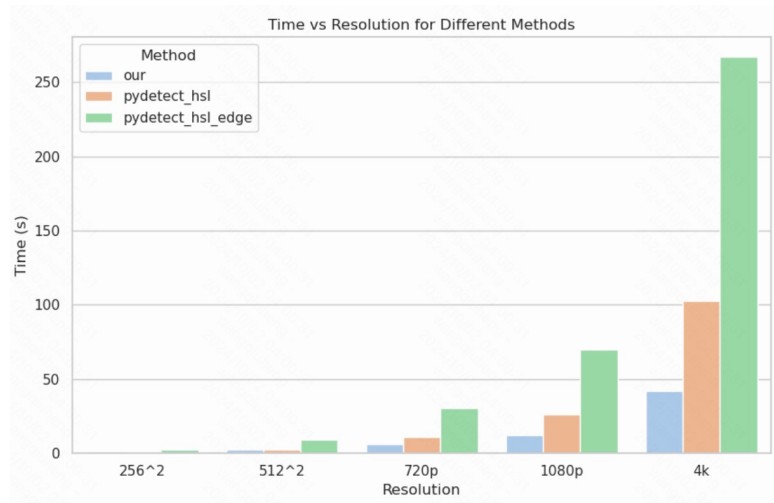

Figure 11: **Time Consumption for Different Resolutions and Methods.** Our method is faster than the others at higher video resolutions.

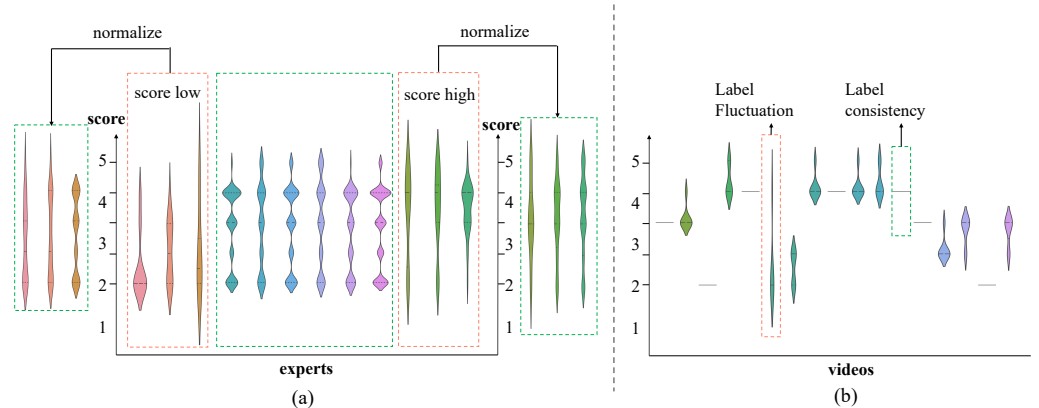

Figure 12: **Score distribution of different experts and videos.** Fig.(a) visualizes the score distribution of different experts. We eliminate individual preference bias through normalization. Fig.(b) visualizes the score distribution of different videos. We reduce label fluctuation bias with average.

## A.3 ABLATION EXPERIMENTS OF TRAINING SUITABILITY ASSESSMENT NETWORK

Table 5: Performance Metrics for Different Combinations of Video, Image, and Feature

| Dynamic branch | Static branch | Feature branch | WCGB | PLCC↑ | SRCC↑ | KRCC↑ | RMSE↓ |
|:---:|:---:|:---:|:---:|:---:|:---:|:---:|:---:|
| ✓ | | | | 0.8684 | 0.8580 | 0.7027 | 0.4644 |
| ✓ | ✓ | | | 0.8730 | 0.8637 | 0.7111 | 0.4555 |
| ✓ | ✓ | ✓ | | 0.8953 | 0.8864 | 0.7397 | 0.4203 |
| ✓ | ✓ | ✓ | ✓ | **0.8974** | **0.8868** | **0.7406** | **0.4099** |

We conduct comprehensive ablation experiments on our Training Suitability Assessment Network. The experimental results are shown in Tab 5 . The baseline model utilizes only dynamic features. Adding the static branch enables the model to capture more static information, thereby improving overall performance. The inclusion of the feature branch allows the model to leverage additional label information, further enhancing its performance. The WCGB module integrates label information

with dynamic and static features through a cross-gating mechanism, achieving optimal performance. Each module addition significantly boosts the model's performance.

Combining dynamic and static branches allows the model to capture both types of information. The feature branch utilizes label information for further improvement. The WCGB module optimizes feature integration, achieving the best results.

## A.4 COMPARISON OF RESULTS FROM DIFFERENT METRICS CONDITIONS

Figure 13: **Comparison of results from different metrics conditions.** Our method has more precise control under the same normalized metrics score and stronger ability to decouple control over different metrics, when the style of videos transfer with the motion score.

## A.5 MORE QUALITATIVE RESULTS OF TEXT-TO-VIDEO GENERATION

Figure 14: **More qualitative results of text-to-video generation.**

