# OpenReview forum: "Kinda-45M: A Large-scale Video Dataset Improving Consistency between Fine-grained Conditions and Video Content"
_ICLR.cc/2025/Conference — ICLR 2025 Conference Withdrawn Submission_

### Official Review · Reviewer_VbFh · 2024-10-29

**Soundness:** 2
**Presentation:** 2
**Contribution:** 2
**Rating:** 5
**Confidence:** 4

**Summary:**

This paper focuses on establishing a pipeline for data curation and annotation tailored for video generation. The authors assert that temporal splitting, detailed captions, and video quality filtering are crucial factors that influence dataset quality. The challenge of constructing and filtering datasets for video generation is critical, and the three questions raised by the authors are highly relevant in the current landscape of video generation research. By training additional classifiers, the authors aim to develop a more accurate automated approach for video splitting and filtering. The method is straightforward, while the effectiveness requires further validation.

**Strengths:**

- The paper proposes a pipeline for data curation and filtering. Considerable effort has been made for curating the Panda-70M dataset down to 45M samples that are more suitable for video generation training.
- The experiments demonstrate the superiority of the curated data compared to the Panda-70M dataset.

**Weaknesses:**

- It seems hard to evaluate whether this dataset can enhance the quality of training for video generation models. The vBench score is 0.74, which is lower than that of existing open-source T2V models, such as VideoCrafter-2.0 (0.80), AnimatedDiff-v2 (0.80), LaVie (0.77), and Latte (0.77).
- The explanations for more precise temporal splitting and filtering seem somewhat straightforward, lacking reasonable and theoretical justification. Additionally, some details are missing. For instance, in temporal splitting, can the constructed method effectively handle gradual transitions and transitions for fast-motion scenes? Regarding the construction of the Video Training Suitability Score, how many videos were evaluated by the experts, and how do you ensure that the evaluation standards are consistent among different experts? How were these experts selected?

**Questions:**

- The score distribution from the eight experts appears to resemble a mixture of two Gaussian distributions. Could you provide an explanation or analysis for this observation?
- Will the data and tools be made publicly available? If so, this would be a significant contribution to the field and industry.
- In the validation set for video splitting, I am interested in the ratio of gradual transitions and transitions for fast-motion scenes, as well as the effectiveness of the proposed method for handling such transitions. Additionally, it would be beneficial to include visual results.
- In conducting the effectiveness experiments in Table 2, I notice that all of the datasets passing through a total of 140M data samples. Does this setup ensure that the model converges sufficiently? As the dataset size increases, the number of epochs per sample becomes smaller. How was the decision to set the data sample at 140M considered?

### Minor：
- The first sentence in line 706 appears to be redundant.

---

### Official Review · Reviewer_iqtm · 2024-11-03

**Soundness:** 3
**Presentation:** 2
**Contribution:** 3
**Rating:** 5
**Confidence:** 3

**Summary:**

This paper introduces Kinda-45M, a large-scale, high-quality video dataset designed to enhance the consistency between fine-grained conditions and video content. The authors argue that the quality of video datasets is critical for the performance of video generation models and identify temporal splitting, detailed captions, and video quality filtering as key factors in dataset quality. The paper presents a refined data processing pipeline that includes a linear classifier for transition detection, structured captions, and a Video Training Suitability Score (VTSS) for filtering high-quality videos. The authors claim that their approach leads to better performance and controllability of video generation models.

**Strengths:**

1. The introduction of the Kinda-45M dataset represents a significant contribution to the domain of video generation. The authors meticulously address critical considerations, including temporal splitting, comprehensive captioning, and rigorous video quality filtering, which are often neglected in other datasets.

2. The proposed data processing pipeline exhibits a well-organized structure, enhancing the consistency between fine-grained conditions and video content. The employment of a linear classifier for transition detection, coupled with the introduction of the Video Training Suitability Score (VTSS) for video filtering, represents innovative and practical solutions.

3. The experiments demonstrate the effectiveness of the Kinda-45M dataset. The benchmark comparison with other datasets (e.g., Panda-70M) clearly shows the advantages of Kinda-45M. The ablation experiment distinctly showcased the efficacy of the re-splitting algorithm, data filtering, and metric conditions.

**Weaknesses:**

1. Although this paper offers a comprehensive overview of the data processing pipeline, certain sections, such as the transition detection method utilizing the Color-Struct SVM and the VTSS computation, could be further elucidated. For example, providing detailed implementation specifics of the SVM classifier and the dynamic/static feature fusion would enhance the reader's understanding and accessibility to the method.

2. At line 285, it is mentioned that structured captions often come with redundant information exceeding 300 words. How to limit the caption to around 200 words? A deeper discussion on the quality control mechanism for generating captions would be beneficial, especially how structured caption systems ensure they do not contain redundant information. What are the specific methods? There is a lack of further discussion here.

3. In the experimental section, although the paper extensively compares the Kinda-45M and Panda-70M datasets, it lacks a more comprehensive comparison with other large-scale video text datasets. Further comparisons would better highlight the value of the work on the Kinda-45M dataset.

**Questions:**

1. Clarify Methodology: Technical details of the transition detection method and the process of VTSS should be supplemented. This will help me better understand the methodology in this paper. The structured captions especially how the structured captioning system ensures that it does not contain redundant information should be further illustrated.

2. Broader dataset comparison: A more comprehensive comparison with other large-scale videotext datasets helps further gain a holistic understanding of the contributions of Kinda-45M.

---

### Official Review · Reviewer_DkRf · 2024-11-03

**Soundness:** 3
**Presentation:** 2
**Contribution:** 2
**Rating:** 5
**Confidence:** 4

**Summary:**

This paper proposes the Kinda-45M dataset for text-to-video generation. Rather than directly applying thresholds to metrics, it introduces networks for metric evaluation. First, a color-structured SVM is trained to identify clip boundaries for video splitting. Next, videos are captioned in a structured format to produce long, descriptive captions. Third, a video training suitability network, trained with human-annotated scores, filters the data. Finally, these metrics are incorporated into the video model’s training to enhance quality by conditioning on the metrics. The proposed dataset demonstrates improved quality over the baseline Panda-45M across various aspects on the comprehensive VBench benchmark.

**Strengths:**

- The use of a structured prompt, encompassing elements like subject, motion, environment, and style, is an interesting approach that could enhance various aspects of caption quality.
- The metric conditioning method contributes valuable fine-grained control over the generated results.
- Table 2 and Figure 9 demonstrate that the video generation model trained on Kinda-45M outperforms the model trained on Panda-45M on VBench.

**Weaknesses:**

- This paper introduces the Kinda-45M dataset, which builds upon Panda-45M but applies different criteria for data splitting and filtering. Although it achieves better performance than the baseline, the improvement is minor, and the dataset lacks scalability for training larger models, which makes its contribution somewhat unclear.
- The paper reports that captions average 200 words. Can video models like OpenSora process such long captions without truncation? Is there any study comparing the effectiveness of longer captions versus shorter ones in improving generation results for the same videos?
- The training details for the Training Suitability Assessment network are missing. How large is the annotation dataset used for training this model? How is the criterion in Section 4.3.1 applied to compute the network’s final prediction score?
- The ablated models in Lines 466-472 are difficult to interpret; the descriptions need more specificity regarding what aspects were ablated. Additionally, the definition of Kinda-46M should be clarified.

**Questions:**

- Are there examples of captions generated with this structured prompt?
- Are there examples where metric conditioning allows for variations in motion scale or aesthetic style in the generated videos?

---

### Official Review · Reviewer_LSkA · 2024-11-04

**Soundness:** 3
**Presentation:** 3
**Contribution:** 2
**Rating:** 3
**Confidence:** 5

**Summary:**

This paper proposed a novel dataset Kinda-45M for text-to-video generation task. It introduced a series of data processing techniques, including transition detection methods, structured caption system, Video Training Suitability Score (VTSS) and metric conditions, to obtain accurate video spliting, detailed captions and higher-quality video content. Experiments demonstrate that training on Kinda-45M is able to achieve better performance compated to previous open-source datasets.

**Strengths:**

1. This work proposed a detailed pipeline on data filtering for video generation tasks.
2. An interesting metric, Video Training Suitability Score (VTSS), is introduced for accurate data filtering.
3. Training video generation models with metric score condition seems to be a novel technique.

**Weaknesses:**

1. The experiment part is weak. The paper proposed several data filtering techniques. It is unclear how different techniques contribute to final results? And it is unknown compared to Panda-70M, which techniques are crucial for the performance of video generation.
2. Pand-70M and Kinda-45M have different numbers of videos, the comparison seems to be unfair.
3. It seems that the model is not well-trained from the results in Fig10. For example, the row "Kinda-45M", the model is unable to generate a panda well comapred to row "Kinda-45M Condition". I doubt it only relates to condition.
4. VTSS seems to be a very important metric for data filtering. However, there is no ablation study to prove the effectiveness of such metric.
5. What is the model size for analyzing the dataset? It lacks detailed introduction about model architecture, training strategy, training time, etc.
6. Will the dataset, and all the filtering tools be released?

**Questions:**

see weaknesses

---

### Note · Authors · 2024-11-13

**Comment:**

Thanks for the time of ACs and reviewers,  we decide to withdrawal our paper.

**Withdrawal Confirmation:**

I have read and agree with the venue's withdrawal policy on behalf of myself and my co-authors.